# Awareness of Medical Radiologic Technologists of Ionizing Radiation and Radiation Protection

**DOI:** 10.3390/ijerph20010497

**Published:** 2022-12-28

**Authors:** Sachiko Yashima, Koichi Chida

**Affiliations:** 1Division of Radiation, Miyagi Cancer Society, Sendai 980-0011, Miyagi, Japan; 2Division of Radiological Disasters and Medical Science, International Research Institute of Disaster Science (IRIDeS), Tohoku University, Sendai 980-8577, Miyagi, Japan; 3Department of Radiological Technology, Tohoku University Graduate School of Medicine, Sendai 980-8575, Miyagi, Japan

**Keywords:** comparison of radiation awareness, disaster medicine, questionnaire survey, training and education, radiation protection, radiologic technologist

## Abstract

Japanese people experienced the Hiroshima and Nagasaki atomic bombings, the Japan Nuclear Fuel Conversion Co. criticality accident, it was found that many human resources are needed to respond to residents’ concerns about disaster exposure in the event of a radiation disaster. Medical radiologic technologists learn about radiation from the time of their training, and are engaged in routine radiographic work, examination explanations, medical exposure counseling, and radiation protection of staff. By learning about nuclear disasters and counseling, we believe they can address residents’ concerns. In order to identify items needed for training, we examined the perceptions of medical radiologic technologists in the case of different specialties, modalities and radiation doses. In 2016, 5 years after the Fukushima Daiichi nuclear power plant accident, we conducted a survey of 57 medical radiologic technologists at two medical facilities with different specialties and work contents to investigate their attitudes toward radiation. 42 participants answered questions regarding sex, age group, presence of children, health effects of radiation exposure, radiation control, generation of X rays by diagnostic X ray equipment, and radiation related units. In a comparison of 38 items other than demographic data, 14 showed no significant differences and 24 showed significant differences. This study found that perceptions of radiation were different among radiology technologists at facilities with different specialties. The survey suggested the possibility of identifying needed training items and providing effective training.

## 1. Introduction

The Organization for Economic Cooperation and Development (OECD) statistics show that the number of X-ray computed tomography (CT) machines in Japan is higher than that in other countries [1]. The United Nations Scientific Committee on the Effects of Atomic Radiation (UNSCEAR) 2008 report states that, on average, the world receives 2.4 mSv of natural radiation per person per year [2]. The UNSCEAR 2020/2021 report reports, on average, an additional 0.57 mSv of exposure related to medical diagnoses in addition to exposure from natural radiation [3]. In contrast, the “Radiation in the Living Environment, 3rd Edition” estimates the average annual per capita natural radiation dose to be 2.1 mSv per year per person in Japan, and the annual radiation dose associated with medical diagnoses to be about 2.6 mSv per year [4]. Therefore, in Japan, medical exposure exceeds exposure to natural radiation, indicating that high levels of medical exposure are remarkable worldwide. Japanese people experienced the Hiroshima and Nagasaki atomic bombings, the Tokaimura nuclear accident (the Japan Nuclear Fuel Conversion Co. criticality accident) [5], and the Fukushima Daiichi Nuclear Power Plant accident [6,7,8,9,10,11,12,13,14,15]; therefore, medical radiation exposure is likely to be of significant concern [16,17]. The Fukushima Daiichi Nuclear Power Plant accident was a nuclear disaster that affected a wide area of eastern Japan, and many people outside of Fukushima Prefecture also suffered from anxiety about disaster exposure. It is thought that an individualized response to disaster exposure anxiety will be effective in reducing anxiety, and many human resources capable of responding to anxiety are needed. Medical radiologic technologists learn about radiation from the time of their training and are engaged in routine radiographic work, examination explanations, medical exposure counseling, and the radiation protection of staff, making a significant contribution to the evaluation of patient-exposure measurements and the optimization of dose reduction [18,19,20,21,22,23,24,25,26]. Although they need to learn about disaster exposure and counseling, they may become personnel who can be responsible for responding to residents’ concerns about disaster exposure in the event of a radiation disaster. They also play an important role in occupational radiation protection in facilities with medical radiation equipment [27,28,29,30,31,32,33,34,35,36,37,38]. Radiation safety measures are necessary for medical radiologic technologists, who are familiar with radiation risks and protection [39,40,41,42]. Staff in medical facilities may have different specialties and job characteristics. According to the report by the Ministry of Health, Labour, and Welfare of Japan, the number of medical radiological technologists has increased from 2002 to 2014 [43], and the number of more specialized and more qualified certified technologists is increasing (i.e., magnetic resonance technological specialists, X-ray CT technologists, radiotherapy technologists, and radiologic technologists for screening mammography), similar to physicians and nurses. In 2016, 5 years after the Fukushima Daiichi Nuclear Power Plant accident, we conducted a survey on ionizing radiation and radiation protection awareness among radiologic technologists at two medical facilities with different specialties and job descriptions. The survey results were reviewed to determine whether perceptions of radiation also differ when the nature of the radiological work and the modalities used, and the radiation doses were significantly different. Since the Fukushima nuclear accident, various surveys have been conducted on the general public, doctors, medical students, and others [44,45,46,47]; however, few surveys have been conducted on medical radiologic technologists. Therefore, in this paper, we retrospectively review and report the survey data.

## 2. Materials and Methods

Fifty-seven medical radiologic technologists from two medical facilities (facility A, facility D) were surveyed using 41 items multiple choice questionnaire; the survey was completed anonymously. Facility A mainly provides cancer screening in the fields of digestive organs, gynecology, and medical checkups. Facility D mainly provides the examination and treatment of the cardiovascular, gastrointestinal, and respiratory organs. Regarding the objective to understand whether medical radiologic technologists’ perceptions on radiation differ when the nature of their radiological work and the modalities and radiation doses they use differ significantly, we conducted a survey on attitudes toward radiation among medical radiologic technologists working at the two facilities with different specialties and daily work routines and compared the results.

The questionnaire items were as follows:GenderAge groupDo you have a family member of elementary school age or younger?Do you think that radiation exposure can cause deformities?If radiation exposure causes deformities, when do you think that exposure causes deformities?Do you think it is better to have an abortion if you are exposed to radiation during pregnancy, for example, during a stomach examination?Do you think your own exposure to radiation will affect your children (ages 0–15)?Do you think health effects occur when children under the age of 15 years are exposed to radiation?Do you believe that the genetic effects occur as a result of radiation exposure?Do you think that radiation exposure can cause cancer or leukemia?Do you think that genetic damage caused by radiation cannot be repaired (healed)?Do you think that genetic damage can be caused by anything other than radiation?If leukemia develops because of radiation exposure, how long do you think it will take to develop after exposure?If thyroid cancer is caused by radiation exposure, how long do you think it will take to develop after exposure?When do you think X-rays come out of the X-ray machine?What do you think happens to the X-rays emitted from an X-ray machine afterwards?Do you think there are health effects from exposure to radiation tests received in hospitals?Do you understand the three principles of external radiation protection?Do you know the half-life of radioactive materials?Do you think that when you are exposed to radiation from a CT scan, etc., that radiation remains in your body forever?Do you think there is exposure to natural radiation during normal times (when there are no nuclear accidents)?Do you think that our bodies also emit radiation?Do you think food or medical devices are ever being irradiated?Do you think that the effects on the body differ between natural and artificial radiation (e.g., medical exposure)?If radiation exposure (an effective dose) is equal, do you think that internal exposure has greater health effects than external exposure?If a deformed baby is born near a nuclear power plant 2 months after the accident, do you suspect that the accident has affected the baby?Do you not think that rice, vegetables, fish, etc. contain very small amounts of naturally occurring radioactive materials?Do you know the differences between radiation and radioactivity?Do you know that there are various types of radiation?Do you know the annual dose limits for radiation workers?Do you check your personal dosimeter reports yourself ?Do you know the content of your personal dosimetry report?Do you think the effects on the body are the same for a single radiation exposure of 100 mSv or for a total of multiple exposures of 100 mSv?Do you think that areas with higher natural radiation levels have higher cancer mortality rates?Do you think there will be health effects if the exposure dose exceeds 1 mSv/year?Do you know the stochastic effects of radiation exposure?Do you know the deterministic effects (tissue reactions) of radiation exposure?Do you know what a becquerel (unit of radiation) is?Do you know what a sievert (unit of radiation) is?Do you know the differences between sievert, millisievert, and micro sievert?Do you find it difficult to understand radiation?

Since the answers to the survey items No. 4, No. 7, No. 8, No. 9, No. 10, No. 11, and No. 17 depend on the dose level of exposure, the rough dose level was used as the answer choice. The survey results were compared and analyzed using Fisher’s exact probability test with the JMP (Ver. 16) (*p* < 0.05).

This study protocol was approved by the Ethics Review Committee of the Miyagi Cancer Society on 8 February 2022 (No. 2108).

## 3. Results

Thirty radiologic technologists from facility A and 27 radiologic technologists from facility D were surveyed; 21 from facility A and 21 from facility D responded to the survey. The response rates were 70.0% for facility A and 77.8% for facility D.

### 3.1. Demographic Data of Respondents

Both facilities had 2–2.5 fold more men than women (Figure 1). The largest age group of respondents in facility A was their 50 s, accounting for one-third of the respondents. Approximately 81% of the respondents at facility D were in their 30 s or younger (Figure 2). Box-and-whisker plots by age group are shown. The p-values from the likelihood ratio and Cochran-Armitage trend test were 0.0374 and 0.0024, respectively (Figure 3. Of the respondents in Facility A, 28.6% (*n* = 6) had children of elementary school age or younger in their family; in facility D, only one respondent had a child of elementary school age or younger (Figure 4).

### 3.2. Questionnaire Items with Matched Response Selection Ratios

The answer selection rates were consistent for Items No. 24 (Figure 5) and No. 41 (Figure 6). Regarding Item No. 24, the health effects of natural or artificial radiation, 23.8% of the respondents in both institutions answered “different”. Regarding Item No. 41, “understanding radiation,” 85.7% of the respondents in both facilities answered “difficult”.

### 3.3. Questionnaire Items for Which No Significant Differences Were Found (Excluding Three Matched Items)

No significant differences were found in survey items No. 8, No. 12, No. 16, No. 18, No. 20, No. 21, No. 23, No. 25, No. 27, No. 29, and No. 33. We present the results of survey items No. 18 (Figure 7), No. 25 (Figure 8), and No. 29 (Figure 9). Although a significant difference was not found, regarding item No. 18, “ Do you understand the three principles of external radiation protection?”, all respondents at facility D answered “Yes”, while at facility A, 81% answered “Yes”, 9.5% answered “I know two of the three principles”, and 9.5% answered “Don’t know”. Regarding item No. 25, “If radiation exposure (effective dose) is equal, do you think that internal exposure has greater health effects than external exposure?”, 61.9% at facility A, 66.7% at facility D, and more than half in both facilities answered “Bigger”. “The same” was 28.6% at facility A and 33.3% at facility D, and 9.5% at facility A answered “Don’t know”. Regarding item No. 29, “Do you know that there are various types of radiation?”, the percentage of respondents who answered “I know” or “I know roughly” was 100% at facility A and 95.2% at facility D, while 4.8% at facility D answered “Vaguely familiar”.

### 3.4. Questionnaire Items for Which Significant Differences Were Found (p < 0.05)

The results of statistical comparisons with significant differences are shown in Table 1. The results for 12 of the 24 items are presented below: (Figure 10, Figure 11, Figure 12, Figure 13, Figure 14, Figure 15, Figure 16, Figure 17, Figure 18, Figure 19, Figure 20, Figure 21, Figure 22 and Figure 23). We present the results of survey items No. 6 (Figure 10), No. 6 (Figure 11), No. 9 (Figure 12), No. 11 (Figure 13), No. 13 (Figure 14), No. 14 (Figure 15), No. 22 (Figure 16), No. 26 (Figure 17), No. 26 (Figure 18), No. 30 (Figure 19), No. 34 (Figure 20), No. 35 (Figure 21), No. 36 (Figure 22), and No. 37 (Figure 23). Regarding item No. 6, “Do you think it is better to have an abortion if you are exposed to radiation during pregnancy, for example, during a stomach examination?”, 19% at facility A answered “I think so a little” and 4.8% answered “I strongly agree”, and 28.6% at facility D answered “Don’t know”. Regarding item No. 26, “If a deformed baby is born near a nuclear power plant 2 months after the accident, do you suspect that the accident has affected the baby?”, the “Not at all” response rate was 71.4% at facility A and 52.4% at facility D.

## 4. Discussion

The Fukushima Daiichi Nuclear Power Plant accident was a nuclear disaster that affected a wide area of eastern Japan, and many people outside of Fukushima Prefecture also suffered from anxiety about disaster exposure. The nuclear accident has shown that a lot of human resources are needed to respond to residents’ concerns about disaster exposure in the event of a radiation disaster. Medical radiologic technologists learn about radiation from the time of their training, and are engaged in routine radiographic work, examination explanations, medical exposure counseling, and radiation protection of staff. By learning about nuclear disasters and counseling, we believe they can address residents’ concerns. In order to identify items needed for training, we examined the perceptions of medical radiologic technologists. After the Fukushima nuclear power plant accident, previous studies surveyed the general public, doctors, nurses, medical students, and others on their awareness of environmental radiation and disaster exposure [8,48,49], as well as those that surveyed medical radiation workers, including medical radiologic technologists, on their awareness of occupational exposure [50,51,52,53,54,55,56]. However, few previous studies have surveyed radiation awareness, including environmental radiation and disaster exposure, among medical radiologic technologists [57]. More specialized and more qualified to certified technologists are increasing, they have developed specialized roles in their work. Our expectation from the survey was to clarify whether there are differences in the perception of radiation depending on the nature of the work and the modalities and radiation doses used. We also wanted to identify the knowledge that is lacking in personnel who would be able to respond to the concerns of residents in the event of a nuclear disaster, and to identify the education and training items that would be necessary. We wanted to show that the training items may be different for each specialized work. The novelty of this study lies in the fact that radiation awareness surveys were conducted among medical radiologic technologists and comparisons were made among medical radiologic technologists with different specialties. In the items where the survey results did not show significant differences, it was acknowledged that an equal number of respondents (28.6%) perceived the health effects of natural and artificial radiation to be different. In facility A, 9.5% answered “Don’t know” about “three basic protective measures in radiation safety”. In both facilities, it was also found that many respondents perceived internal exposure to be more harmful than external exposure at radiation doses of an equal effective dose. We believe that it is necessary to take measures to ensure a correct understanding of radiation, such as the fact that the health effects of both natural and artificial radiation are the same, and that when the effective dose is equal, the health effects of external exposure and internal exposure are the same, and three basic protective measures in radiation safety. These are the items for which we found significant differences in responses. Item No. 6: If a gastric X-ray examination was performed during pregnancy, 23.8% of the respondents at facility A answered that they would “prefer to have an abortion” or “I think so a little”. In facility D, 28.6% of the respondents were unsure. Item No. 26: A total of 23.8% from facility A and 38.1% from facility D suspected radiation effects if deformities were found in their baby 2 months after the nuclear accident. For items 6 and 26, we also conducted an analysis by age, comparing the responses of those in their 30s and younger with those in their 40s and older, and found significant differences. The percentage of respondents who were misinformed was higher among those aged 30s and younger than among those aged 40s and older, we believe that there is a need to educate those aged 30s and younger on the effects of radiation on the fetus. In a previous study of nurses, <10% answered correctly regarding the effects of radiation exposure during pregnancy on the fetus [48]. We believe that it is also necessary to make people aware of the effects of radiation on fetuses. Only two respondents from facility D stated that there was no genetic effect of radiation. We believe that it is necessary to inform the public that no genetic effects of radiation on humans have been observed to date. For cancer mortality in areas with high Natural radiation levels, at facility A, 19% responded “Slightly high" and 9.5% responded “Don’t know”. At facility D, 4.8% responded "Slightly high” and 38.1% responded “Don’t know”. Facility D showed a higher percentage than Facility A in recognizing a shorter latency period to develop leukemia and solid tumors due to radiation exposure than the medically known period. Facility A, we admitted that the percentage of “I know” responses for “three basic protective measures in radiation safety”, “radiation control”, “probabilistic effects”, “deterministic effects”, and “radiation units” was lower than that of Facility D. A comparative study of perceptions of radiation among medical radiologic technologists at two medical facilities with different specialties, modalities and radiation doses used showed that the level of recognition varied depending on the survey items. The survey suggested the possibility of identifying needed training items and providing effective training and education. Medical radiologic technologists play an important role in managing the exposure of patients to ionizing radiation [58,59]. This role is to optimize the medical exposure of patients, striving to keep the dose below that at which definite effects occur and keeping the incidence of stochastic effects as low as possible [60]. In Japan, the first diagnostic reference level was presented in 2015, and awareness of radiation protection among medical radiation workers seems to have increased [61,62,63]. Furthermore, in addition to managing patient radiation exposure, medical radiologic technologists play an important role in managing occupational exposure protection [64,65,66,67,68,69,70]. We consider that it is essential for radiation safety that medical radiologic technologists are well-versed in radiation protection. It is necessary to understand the status of radiation awareness among medical radiologic technicians and to provide effective education on items that are not correctly recognized or are difficult to understand. On 1 April 2021, Japan’s Enforcement Regulations on the Medical Care Act were amended and came into effect. The following measures are specified to ensure a system for the safe management of medical radiation: (a) Formulate guidelines for the safe use of medical radiation. (b) Provide training for the safe use of medical radiation to persons engaged in radiation treatment. (c) Undertake improvement measures for the management and recording of the radiation exposure dose of persons who receive radiation treatment using specific items and for the safe use of medical radiation. We believe that the need to disseminate the knowledge of radiation protection necessary to deal with examinees and medical personnel through staff training and education will increase further in the future.

## 5. Conclusions

The nuclear accident show that a large number of human resources were needed to deal with residents’ concerns about disaster exposure in the event of a radiation disaster. We believe that medical radiologic technologists learning on nuclear disasters and counseling will help them respond to residents’ concerns. We wanted to identify the knowledge lacking in responding to residents’ concerns in the event of a nuclear disaster, to show that the training items may be different for each specialized work. In 2016, a survey of medical radiologic technologists at two medical facilities with different practices and specialties was conducted to determine their attitudes toward radiation; the survey results were retrospectively compared. In a comparison of 38 items other than demographic data, 14 showed no significant differences, and 24 showed significant differences. It was acknowledged that an equal number of respondents perceived that the health effects of natural and artificial radiation to be different. In both facilities, it was also found that many respondents perceived internal exposure to be more harmful than external exposure at radiation doses of equal effective dose. The study on the effects of exposure on the fetus during pregnancy by age group showed that there was low awareness among those in their 30s and younger. We conducted the comparative study on the perceptions among medical radiologic technologists of ionizing radiation and radiation protection at two facilities with different specialties, modalities and radiation doses used, and they were found to be different depending on the survey items. The survey suggested the possibility of identifying needed training items and providing effective training and education.

## Figures and Tables

**Figure 1 ijerph-20-00497-f001:**
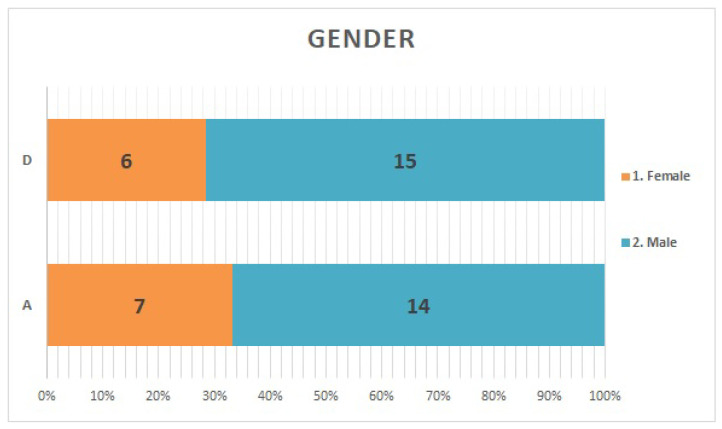
Gender: Both facilities had more than twice as many men as women.

**Figure 2 ijerph-20-00497-f002:**
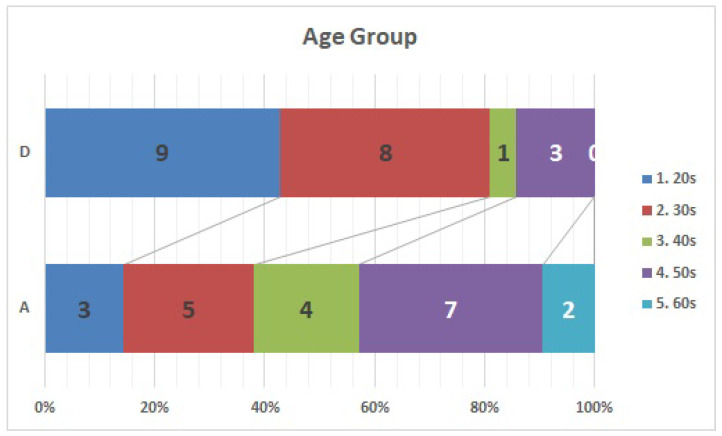
Age Group: Respondents in facility D tended to be younger than those in facility A.

**Figure 3 ijerph-20-00497-f003:**
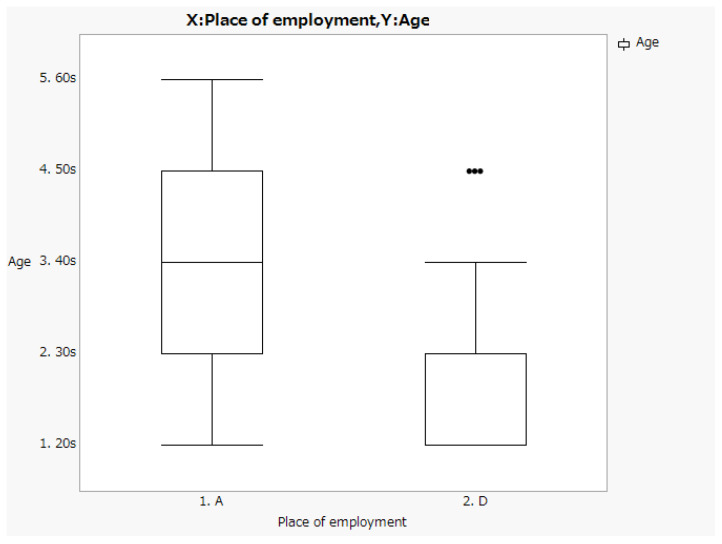
Box-and-whisker plots by age group.

**Figure 4 ijerph-20-00497-f004:**
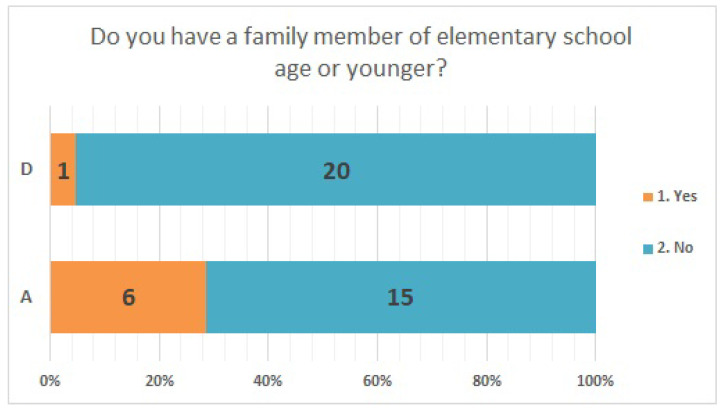
Presence of children of elementary school age or younger.

**Figure 5 ijerph-20-00497-f005:**
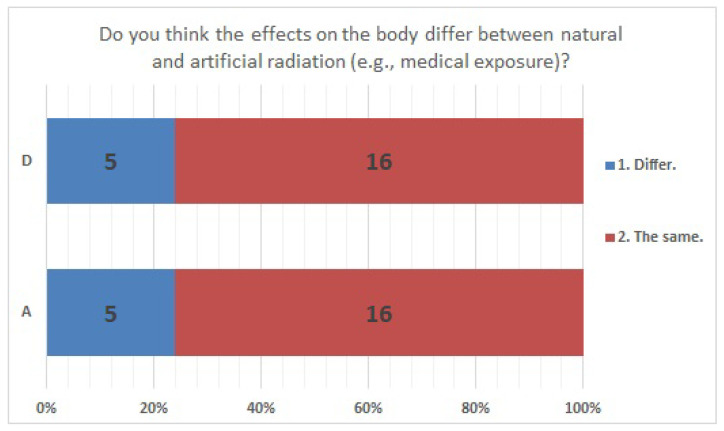
Responses to item No. 24.

**Figure 6 ijerph-20-00497-f006:**
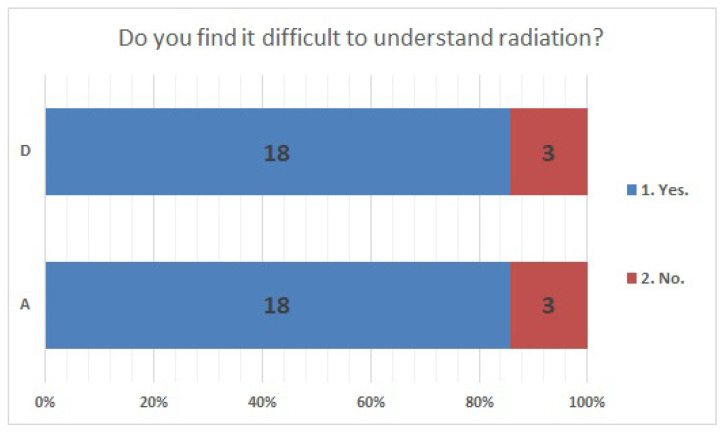
Responses to item No. 41.

**Figure 7 ijerph-20-00497-f007:**
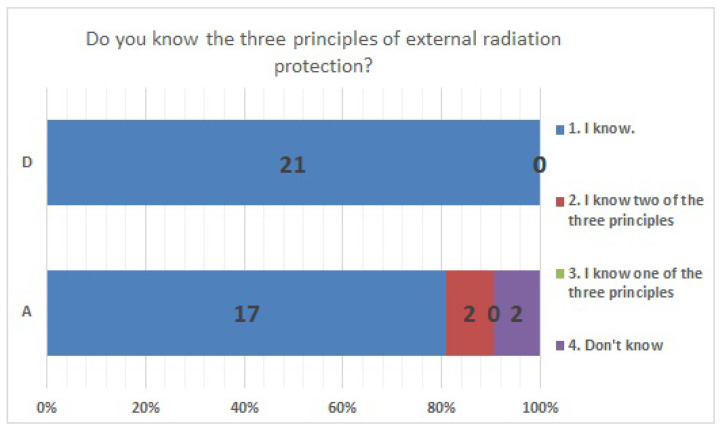
Responses to item No. 18.

**Figure 8 ijerph-20-00497-f008:**
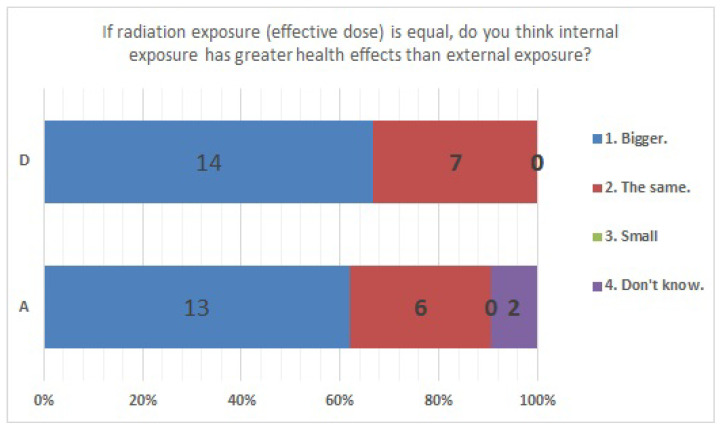
Responses to item No. 25.

**Figure 9 ijerph-20-00497-f009:**
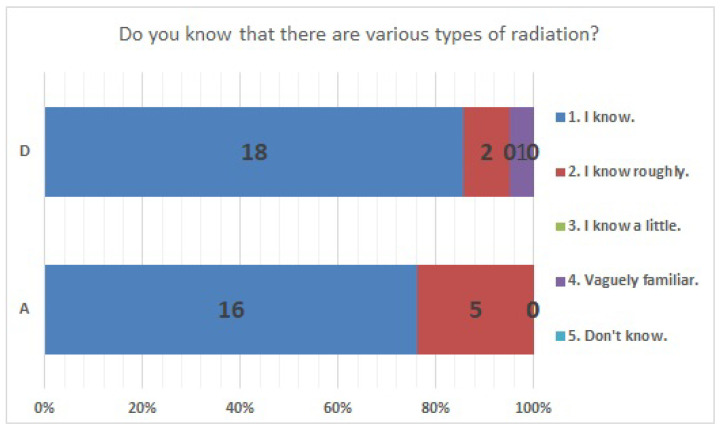
Responses to item No. 29.

**Figure 10 ijerph-20-00497-f010:**
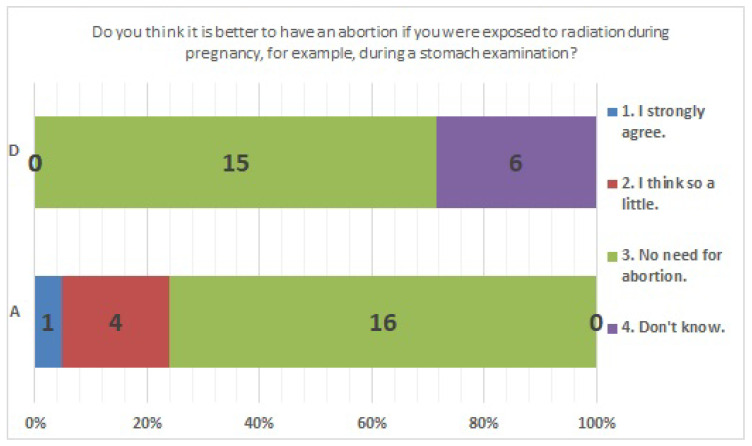
Responses to item No. 6.

**Figure 11 ijerph-20-00497-f011:**
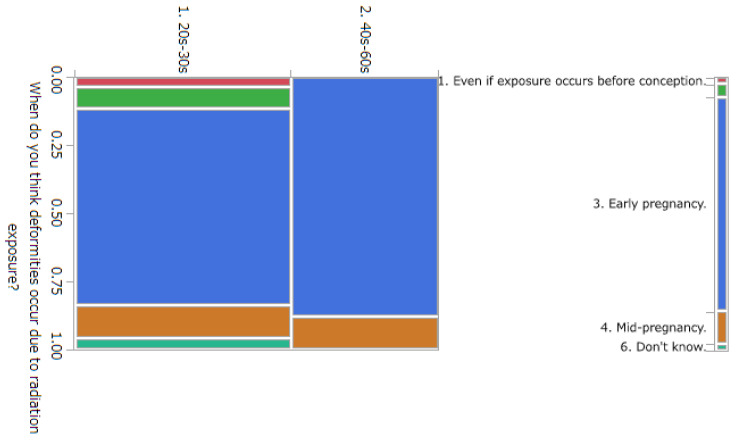
Responses to item No. 6 by Age Group.

**Figure 12 ijerph-20-00497-f012:**
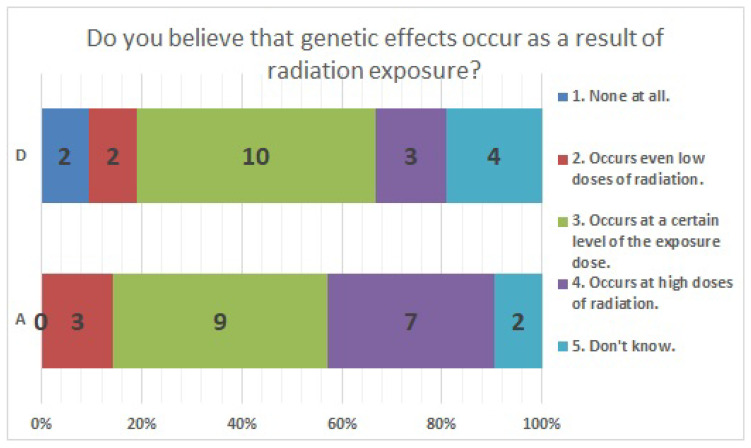
Responses to item No. 9.

**Figure 13 ijerph-20-00497-f013:**
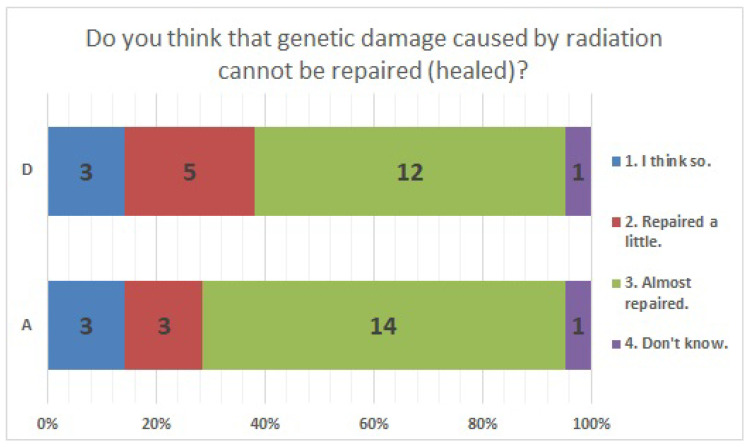
Responses to item No. 11.

**Figure 14 ijerph-20-00497-f014:**
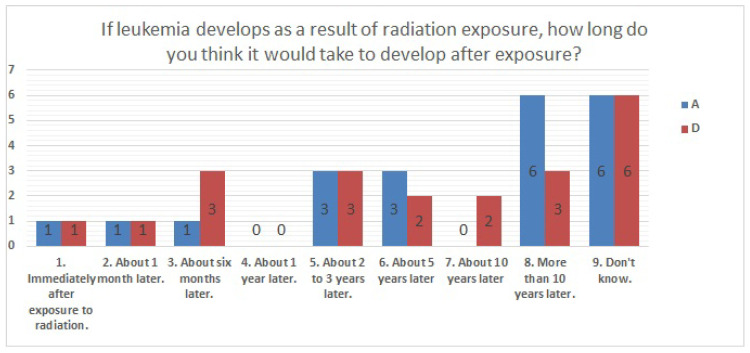
Responses to item No. 13.

**Figure 15 ijerph-20-00497-f015:**
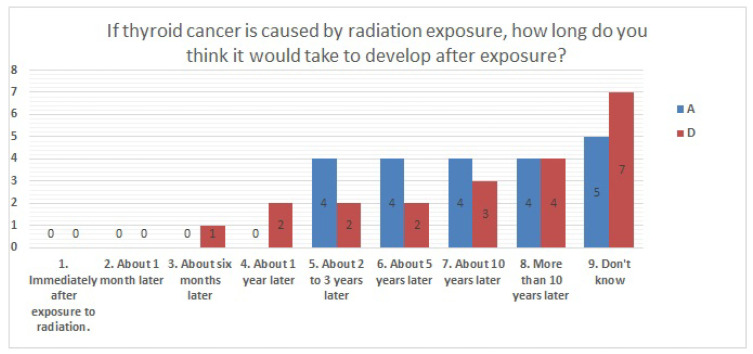
Responses to item No. 14.

**Figure 16 ijerph-20-00497-f016:**
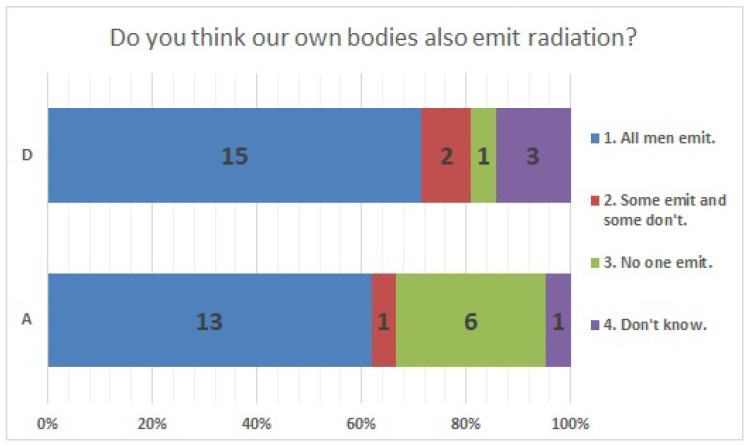
Responses to item No. 22.

**Figure 17 ijerph-20-00497-f017:**
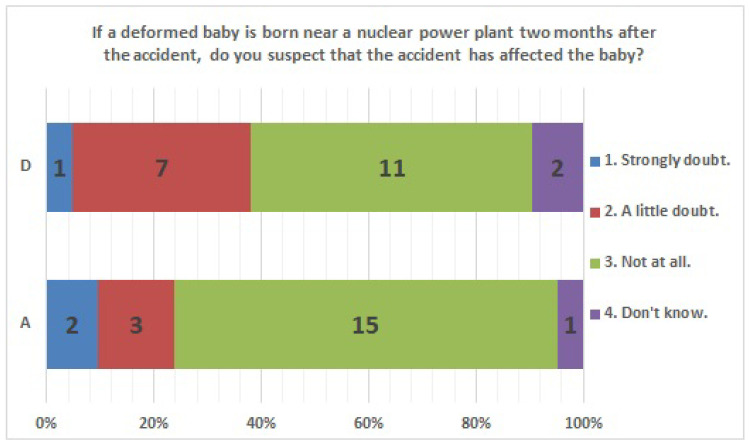
Responses to item No. 26.

**Figure 18 ijerph-20-00497-f018:**
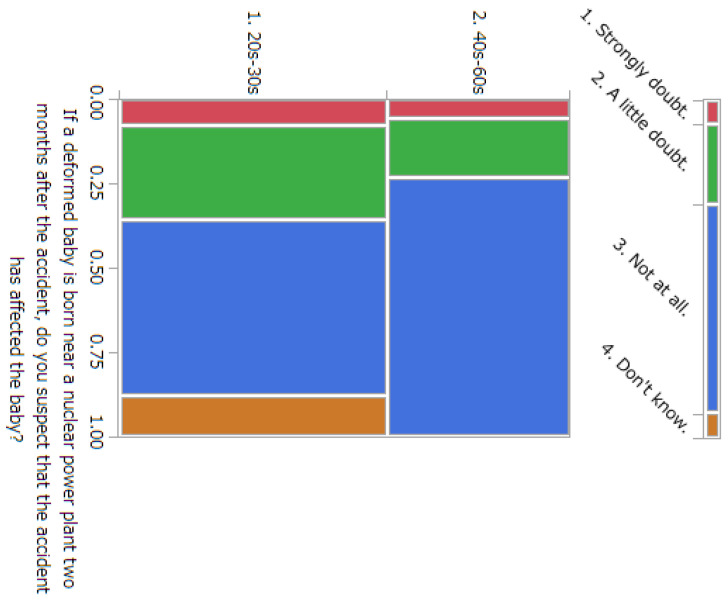
Responses to item No. 26 by Age Group.

**Figure 19 ijerph-20-00497-f019:**
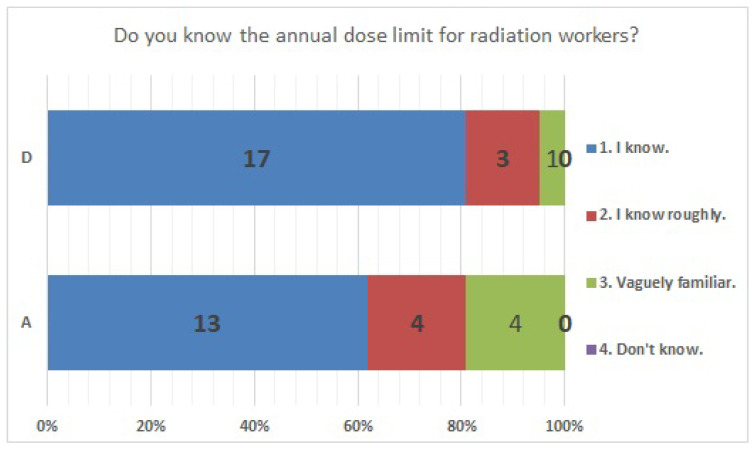
Responses to item No. 30.

**Figure 20 ijerph-20-00497-f020:**
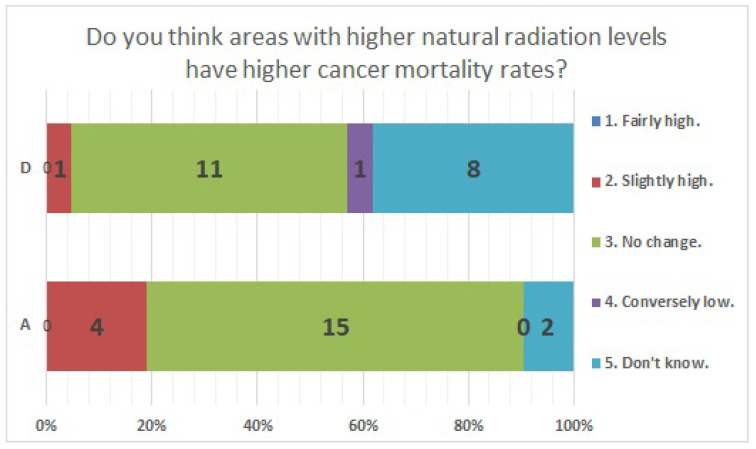
Responses to item No. 34.

**Figure 21 ijerph-20-00497-f021:**
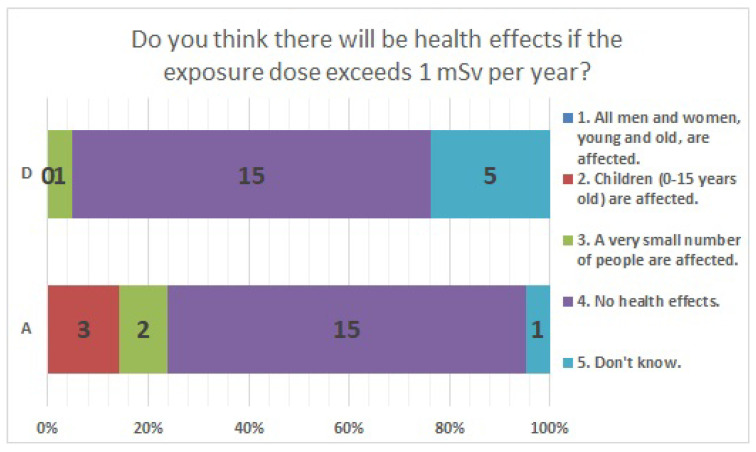
Responses to item No. 35.

**Figure 22 ijerph-20-00497-f022:**
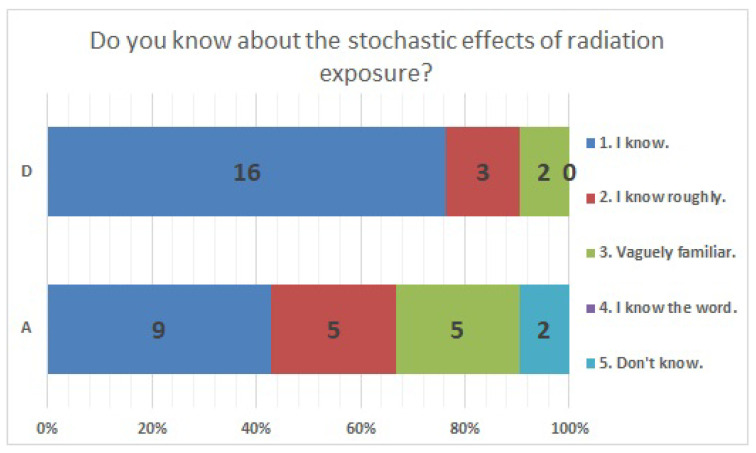
Responses to item No. 36.

**Figure 23 ijerph-20-00497-f023:**
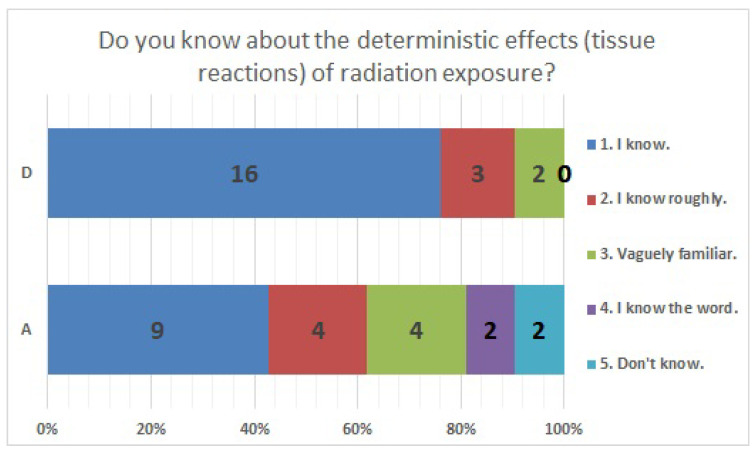
Responses to item No. 37.

**Table 1 ijerph-20-00497-t001:** Questionnaire item number for significant differences and *p* value.

Item No.	*p*-Value	Item No.	*p*-Value
4	0.028	26	0.016
5	0.0006	28	0.015
6	0.0005	30	0.039
7	0.013	31	0.036
9	0.003	32	0.013
10	0.04	34	0.003
11	0.04	35	0.005
13	0.0005	36	0.004
14	0.0008	37	0.002
17	0.037	38	0.012
19	0.033	39	0.01
22	0.0058	40	0.024

## Data Availability

Not applicable.

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
