# Peer review of "Awareness of Medical Radiologic Technologists of Ionizing Radiation and Radiation Protection"

_ijerph, 2022, doi:10.3390/ijerph20010497_

Round 1
Reviewer 1 Report
This study reports the awareness of radiological technologists in Japan. The contents includes useful information in the field of radiological protection. However, there are unclear points and insufficient descriptions in the manuscript and the contents are not well-described. They should be revised.
1. Introduction
P1 L20-21
The medical exposure data in the UNSCEAR 2008 report is referred. However, the UNSCEAR 2020/2021 reopert have been published, so it would be better to refer the latest data of the report.
P2 L37
The authors wrote that the number of certified professional technologists has increased. Is this only recent only tendency? For example, according to the report by the Ministry of Health, Labour and Welfare of Japan, the number of radiological technologists has increased during 2002 to 2014.
(https://www.mhlw.go.jp/file/05-Shingikai-10801000-Iseikyoku-Soumuka/0000200803.pdf)
More clear description would be necessary.
2. Materials and Methods
The differences of the facilities between the facility A and D were described with the results of the statistical analyses, however, it is not clear that whether there are differences of the characteristics between the technologists in A and D except the questions on the issue of gender, age and children. If the differences exist, such as having special educations or trainings, they should be described. If not, the meaning to compare the results of A and D would be only the comparisons on the gender, age and children issues. The purpose of the comparisons and analyses should be clearly described in Materials and Methods, and the results should be analyzed in Discussion.
P2L84-
There are questions which the responders could not easily answered. For example, question number 4, 7, 8, 9, 10, 11, 17. The answers would depend on the dose levels of the exposures. How the questions were selected and the validity should be explained.
4. Discussion
There are many general comments and are not enough discussion based on the results of this study. The contents of the Discussion should be results and previous study related, and the possible reasons of the results should also be described.
P12 L204
It would be better for readers’ understandings to describe that the first DRLs were made by medical radiation or radiation protection related organizations, and the revised Japan DRLs 2020 have been incorporated into the regulations for the first time in Japan.
5. Conclusions
The conclusions should be more results-based.
Author Response
We wish to express our strong appreciation to the Reviewer for the thoughtful and constructive feedback you provided regarding our manuscript, Awareness of Medical Radiologic Technologists of Ionizing Radiation and Radiation Protection. We agree with you that suggestion and we have amended.
Point 1: This study reports the awareness of radiological technologists in Japan. The contents include useful information in the field of radiological protection. However, there are unclear points and insufficient descriptions in the manuscript and the contents are not well-described. They should be revised.
Response 1: We have provided additional statements are unclear or inadequate.
Point 2: P1 L20-21; The medical exposure data in the UNSCEAR 2008 report is referred. However, the UNSCEAR 2020/2021 report have been published, so it would be better to refer the latest data of the report.
Response 2: We have referred to the UNSCEAR 2020/2021 edition.
Point 3: P2 L37; The authors wrote that the number of certified professional technologists has increased. Is this only recent only tendency? For example, according to the report by the Ministry of Health, Labour and Welfare of Japan, the number of radiological technologists has increased during 2002 to 2014.
(https://www.mhlw.go.jp/file/05-Shingikai-10801000-Iseikyoku-Soumuka/0000200803.pdf) More clear description would be necessary.
Response 3: We have described it more clearly according to your suggestion.
Point 4: The differences of the facilities between the facility A and D were described with the results of the statistical analyses, however, it is not clear that whether there are differences of the characteristics between the technologists in A and D except the questions on the issue of gender, age and children. If the differences exist, such as having special educations or trainings, they should be described. If not, the meaning to compare the results of A and D would be only the comparisons on the gender, age and children issues. The purpose of the comparisons and analyses should be clearly described in Materials and Methods, and the results should be analyzed in Discussion.
Response 4: We have described the differences. We have added a comparative analysis objective to the method.
Point 5: P2L84-; There are questions which the responders could not easily answered. For example, question number 4, 7, 8, 9, 10, 11, 17. The answers would depend on the dose levels of the exposures. How the questions were selected and the validity should be explained.
Response 5: We have described how we selected the questionnaire items.
Point 6: There are many general comments and are not enough discussion based on the results of this study. The contents of the Discussion should be results and previous study related, and the possible reasons of the results should also be described.
Response 6: We have described possible reasons for the results and provided a discussion of the results and previous studies.
Point 7: P12 L204; It would be better for readers’ understandings to describe that the first DRLs were made by medical radiation or radiation protection related organizations, and the revised Japan DRLs 2020 have been incorporated into the regulations for the first time in Japan.
Response 7: We have changed the description as noted.
Point 8: The conclusions should be more results-based.
Response 8: We have described possible reasons for the results and provided a discussion of the results and previous studies.
We will upload the paper today up to the point where we have made the corrections. We are in the process of revising it now. We will complete the corrections as soon as possible and upload the paper again.

Reviewer 2 Report
The investigation is interesting, although the aim of the investigation is not clear.
It is not clear why the profession of radiological technician in particular was chosen for the analysis?
Please explain what you expected from this investigation.
You compared two groups of people with the same occupation in two different medical faciliates. What did you want to show with this?
Wouldn't it be better (more interesting) if you compared two different occupations?
In addition, the groups you compared do not have the same characteristics. First of all, the respondents in both groups differ significantly in age. Maybe this is one of the reasons why you got a significant difference in the answers to some questions? Please explain.
The paper is well written but needs to be improved to be suitable for publication in the journal.
Special comments:
1. Introduction
Please explain in more detail what the aim of this research was and why you chose two groups of the same occupations in two different hospitals. What were you trying to prove?
2. Material and Methods
Why did you always choose the same statistical test (Fisher's exact) to compare the results of the answers to the questions in the questionnaire? The questions (and the data obtained) are different and sometimes require different tests for comparison. For example, you did not choose a good test for comparing age in two groups. Since age is quantitative data, it would make more sense if you chose a different test.
3. Results
Check whether there is a statistically significant difference in the age of the respondents in the two groups.
All graphs are made in the same way. For some questions it would be better to choose a different way of presenting the results (e.g. the answers to item no. 13 and 14).
Page 5. figure 4. - Figure 4 is redundant. All respondents in both groups gave the same answers. You can write this in the text, but the figure is superfluous because it shows nothing additional.
Pg. 5, Figure 5. The comment on the figure: "24 percent of the medical radiologic technologists at both facilities reported that the effects on the human body differ between natural and artificial radiation" should not be in the Figure caption, but should be written in the disscusion of the paper.
Pg. 7, capter 3.4., line 150-154, It would be better if you wrote all the results of the statistical comparison in a table instead of this sentence: „Significant differences were found in survey items No.4(p=0.028), No.5(p=0.011), 150 No.6(p=0.0005), No.7(p=0.013), No.9(p=0.003), No.10(p=0.04), No.11(p=0.04), No.13(p=0.0005),151 No.14(p=0.0008), No.17(p=0.037), No.19(p=0.033), No.22(p=0.0058), No.26(p=0.016), No.28(p=0.015),152 No.30(p=0.039), No.31(p=0.036), No.32(p=0.013), No.34(p=0.003), No.35(p=0.005), No.36(p=0.004), 153 No.37(p=0.002), No.38(p=0.012), No.39(p=0.01), and No.40(p=0.024).“
Pg. 7-11. Figure 7-21. The comment on the figures 7-21 should not be written in the Figure captions, but should be written in the disscusion of the paper.
1. Conclusion
The conclusion is too short. It does not highlight what conclusions were drawn from this study. What are the differences between medical radiologic technologists in two medical institutions and why do you think they arose? Complete and correct the content of the Conclusion chapter and complete the abstract of the paper accordingly.
Author Response
We wish to express our strong appreciation to the Reviewer for the thoughtful and constructive feedback you provided regarding our manuscript, Awareness of Medical Radiologic Technologists of Ionizing Radiation and Radiation Protection. We agree with you that suggestion and we have amended.
Point 1: The investigation is interesting, although the aim of the investigation is not clear.
It is not clear why the profession of radiological technician in particular was chosen for the analysis?
Please explain what you expected from this investigation.
You compared two groups of people with the same occupation in two different medical faciliates. What did you want to show with this?
Response 1: Our expectation from the survey was to clarify whether there are differences in the perception of radiation depending on the nature of the work and the modalities and radiation doses used. We also wanted to identify the knowledge that is lacking in personnel who would be able to respond to the concerns of residents in the event of a nuclear disaster, and to identify the education and training items that would be necessary. We wanted to show that the training items may be different for each specialized work.
Point 2: Wouldn't it be better (more interesting) if you compared two different occupations?
Response 2: In this study, we compared radiologic technologists with different specialties in order to clarify the differences in perceptions of radiation among radiologic technologists.
Point 3: In addition, the groups you compared do not have the same characteristics. First of all, the respondents in both groups differ significantly in age. Maybe this is one of the reasons why you got a significant difference in the answers to some questions? Please explain.
Response 3: We would like to do an analysis by age, as there may be a bias based on age.
Point 4: Please explain in more detail what the aim of this research was and why you chose two groups of the same occupations in two different hospitals. What were you trying to prove?
Response 4: We have described in more detail the objectives of this study.
Point 5: Why did you always choose the same statistical test (Fisher's exact) to compare the results of the answers to the questions in the questionnaire? The questions (and the data obtained) are different and sometimes require different tests for comparison. For example, you did not choose a good test for comparing age in two groups. Since age is quantitative data, it would make more sense if you chose a different test.
Response 5: We compared age groups, not age, nominal categorical data. We chose Fisher's exact as our test method because the average amount of data for the age group was less than 5, which could have caused problems with the likelihood ratio and chi-square.
Point 6: Check whether there is a statistically significant difference in the age of the respondents in the two groups.
Response 6: We have checked whether there is a statistically significant difference in the age group of the respondents in the two groups.
Point 7: All graphs are made in the same way. For some questions it would be better to choose a different way of presenting the results (e.g. the answers to item no. 13 and 14).
Response 7: We are currently working on a suitable presentation for the answers to item no. 13 and 14.
Point 8: Page 5. figure 4. - Figure 4 is redundant. All respondents in both groups gave the same answers. You can write this in the text, but the figure is superfluous because it shows nothing additional.
Response 8: We removed Figure 4.
Point 9: Pg. 5, Figure 5. The comment on the figure: "24 percent of the medical radiologic technologists at both facilities reported that the effects on the human body differ between natural and artificial radiation" should not be in the Figure caption, but should be written in the discussion of the paper.
Response 9: We have included that comment in the discussion of the paper.
Point 10: Pg. 7, capter 3.4., line 150-154, It would be better if you wrote all the results of the statistical comparison in a table instead of this sentence: „Significant differences were found in survey items No.4(p=0.028), No.5(p=0.011), No.6(p=0.0005), No.7(p=0.013), No.9(p=0.003), No.10(p=0.04), No.11(p=0.04), No.13(p=0.0005), No.14(p=0.0008), No.17(p=0.037), No.19(p=0.033), No.22(p=0.0058), No.26(p=0.016), No.28(p=0.015), No.30(p=0.039), No.31(p=0.036), No.32(p=0.013), No.34(p=0.003), No.35(p=0.005), No.36(p=0.004), No.37(p=0.002), No.38(p=0.012), No.39(p=0.01), and No.40(p=0.024).“
Response 10: We have prepared a table of the results of the statistical comparisons and wrote it in this paper.
Point 11: Pg. 7-11. Figure 7-21. The comment on the figures 7-21 should not be written in the Figure captions, but should be written in the discussion of the paper.
Response 11: We have included that comment in the discussion of the paper.
Point 12: The conclusion is too short. It does not highlight what conclusions were drawn from this study. What are the differences between medical radiologic technologists in two medical institutions and why do you think they arose? Complete and correct the content of the Conclusion chapter and complete the abstract of the paper accordingly.
Response 12: We would like to revise our conclusions and amend our abstract in light of your comments.
We will upload the paper today up to the point where we have made the corrections. We are in the process of revising it now. We will complete the corrections as soon as possible and upload the paper again.

Round 2
Reviewer 2 Report
The authors have responded to all comments and substantially improved the manuscript so that it now meets the requirements for publication.